# Effects of Different InGaN/GaN Electron Emission Layers/Interlayers on Performance of a UV-A LED

**Dohyun Kim [1,†], Keun Man Song [2,†], UiJin Jung [1], Subin Kim [1], Dong Su Shin [1] and Jinsub Park [1,3,\*]**

1   Department of Electronics and Computer Engineering, Hanyang University, Seoul 04763, Korea; dohyunkim@hanyang.ac.kr (D.K.); qwpoiu1002@hanayng.ac.kr (U.J.); subinkim@hanyang.ac.kr (S.K.); naenora@hanyang.ac.kr (D.S.S.)
2   Korea Advanced Nano Fab Center, Suwon, Gyeonggi 16229, Korea; keunman.song@kanc.re.kr
3   Department of Electronic Engineering, Hanyang University, Seoul 04763, Korea
\*   Correspondence: jinsubpark@hanyang.ac.kr
†   These authors equally contributed to this work.

**Abstract:** In this study, we investigated the effects of InGaN/GaN-based interlayer (IL) and electron emitting layer (EEL) consisting of a GaN barrier layer grown with different metal-organic (MO) precursors of gallium (Ga), which were grown underneath the active layer. The growth behavior of GaN with triethyl Ga (TEGa) showed an increasing growth time due to a lower growth rate compared with GaN grown with trimethyl Ga (TMGa), resulting in the formation of columnar domains and grain boundary with reduced defect. UV-A light emitting diode (LED) chips with three types of ILs and EELs, grown with different MO sources, were fabricated and evaluated by light output power (LOP) measurements. The LOP intensity of UVLED-III with the GaN barrier layer-based IL and EEL grown by TEGa was enhanced by 1.5 times compared to that of the IL and EEL grown with TMGa at 300 mA current injection. Use of the GaN barrier layer in ILs and EELs grown by TEGa improved the crystal quality of the post grown InGaN/GaN multiple quantum well, which reduces leakage current. Therefore, for the UV-A LED with ILs and EELs grown with TEGa MO precursors, electrical and optical properties were improved significantly.

**Keywords:** UV-LED; Interlayer; GaN; MOCVD; Electron emission layer

## 1. Introduction

Recently, in addition to the dramatic development of III-nitride visible light wavelength light-emitting diode (LED) regions over the past two decades [1,2], (Al)GaN-based ultraviolet light-emitting diodes (UV LEDs) have attracted considerable attention from many researchers for their versatile applications, such as lighting source in UV photolithography, epoxy-curing, water purification, and in medical heath care agent detection/analysis systems [3]. Despite the many unique applications using UV LEDs compared with blue and green LEDs, they still have problems for use in commercialized industrial fields. These issues arise from the relatively lower quantum efficiency compared to the visible lighting sources that result from the difficulty in obtaining high quality (Al) GaN-based epi-structures, which are an essential active layer for deep UV LEDs. In the UV-A (wavelength range: 315 nm–400 nm) LED, the well/barrier materials are very similar to those of the visible GaN-based LED, but they have relatively less indium (In) in the active region than do blue or green LEDs. A decrease of In contents leads not only to weak carrier localization, but also a small band gap discontinuity at the interface between InGaN well and GaN barrier [4], resulting in decreased radiating efficiency in UV-A LEDs. Generally, to improve crystal quality of epi-structures

in a UV LED, many effective methods have been suggested, including various designs for patterned substrates, super-lattices, interlayers (ILs), and electron emitting layers (EELs) structures [5,6]. Among the suggested methods, the InGaN/GaN-based IL and EEL structures produced before the growth of multiple quantum wells (MQWs) showed important effects to improve the performance of GaN-based optoelectronic devices [7,8]. The increase of electron capture in the EELs and the effective supply of the captured electrons to the well layers through tunneling via the barrier layer are possible mechanisms for improvement of quantum efficiency. Through this process, a large electron capture rate and high carrier confinement are accomplished using the ILs and EELs under the active layers [9].

In this study, we investigated the effects of GaN/InGaN ILs and EELs consisting of a GaN barrier layer grown with the metal-organic (MO) gallium (Ga) precursors of trimethylgallium (TMGa) and triethylgallium (TEGa). The TMGa and TEGa sources generally used for growth of the GaN layer exhibited varied growth mechanisms with different temperature decomposition, diffusion length, and carbon contamination, which affect crystal quality and device performance [10]. Different growth behaviors of GaN barriers in ILs and EELs with change of MO source effect the structural and optical properties of the post-grown active layer resulting in device performance of GaN-based UV-A LEDs.

## 2. Experimental

We grew the UV-A LED epi-structures on a 2-in, c-plane (0001) $Al_2O_3$ (sapphire) substrate using an Aixtron 2600 G3HT metal organic chemical vapor deposition (MOCVD) system (Aixtron, Herzogenrath, Aachen, Germany) [11]. The epi-structures consist of a 200-nm thick GaN buffer layer, a 4-µm thick unintentionally doped GaN layer, a 3-µm thick Si-doped GaN layer, 3 pairs of 2–6-nm thick Si-doped $In_{0.01}Ga_{0.99}N$/GaN interlayers (ILs), 3 pairs of 3–12-nm-thick $In_{0.01}Ga_{0.99}N$/GaN EELs, 6 periods of InGaN/GaN/AlGaN MGWs, and 150-nm thick Mg-doped GaN layer. During the growth of layers, various MO sources such as trimethylaluminum (TMAl), TMGa, TEGa, and trimethylindium (TMIn) were used. To investigate the effects of growth conditions of the EELs and ILs on crystallinity of the post grown active layer and light efficiency of UV-A LEDs, the MO Ga precursors were changed for the barrier GaN layer in ILs and EELs as denoted in Figure 1a. We prepared three structure of full epi-structures (p-GaN/MQWs/EELs/ILs/n-GaN/u-GaN on $Al_2O_3$ substrate) of UV LEDs consisting of different EELs/ILs with GaN barrier layers with TMGa/TMGa (UVLED I), TEGa/TMGa (UVLED II), and TEGa/TEGa (UVLED III), respectively. In addition, GaN barrier layer-terminated EELs/ILs/n-GaN/u-GaN on $Al_2O_3$ substrate were named following the UVLED chips as EI-UVLED I, EI-UVLED II, and EI-UVLED III, respectively. The strain status and structural characteristics of the UVLED wafers were evaluated by high-resolution X-ray diffraction (HR-XRD). The details for UVLED epitaxy structure were investigated by using a high-resolution transmission electron microscopy (HR-TEM) to elucidate the structural characteristics of the InGaN/GaN MQWs. The electroluminescence (EL) of the fabricated UV-A LEDs was investigated using the probe station WPS3100 (Opto system). The photoluminescence (PL) properties of MQWs was measured by a Nd:YAG 266 laser as the excitation source (output power is fixed at 2.0 mW). For LED chip fabrication, general photolithography, patterning, and metallization processes were conducted as follows [12]. The as-grown LED epi-structures/$Al_2O_3$ were partially etched by inductively coupled plasma until the n-type GaN layer was opened. Then a 200-nm thick transparent conductive indium tin oxide (ITO) layer was deposited by electron beam evaporation (ULTECH-UEE). Cr/Au electrodes (10/500 nm) were subsequently evaporated for n-and p-type electrodes. Finally, the UVLED wafer was divided into rectangular chips with an area of $485 \times 425$ µm$^2$.

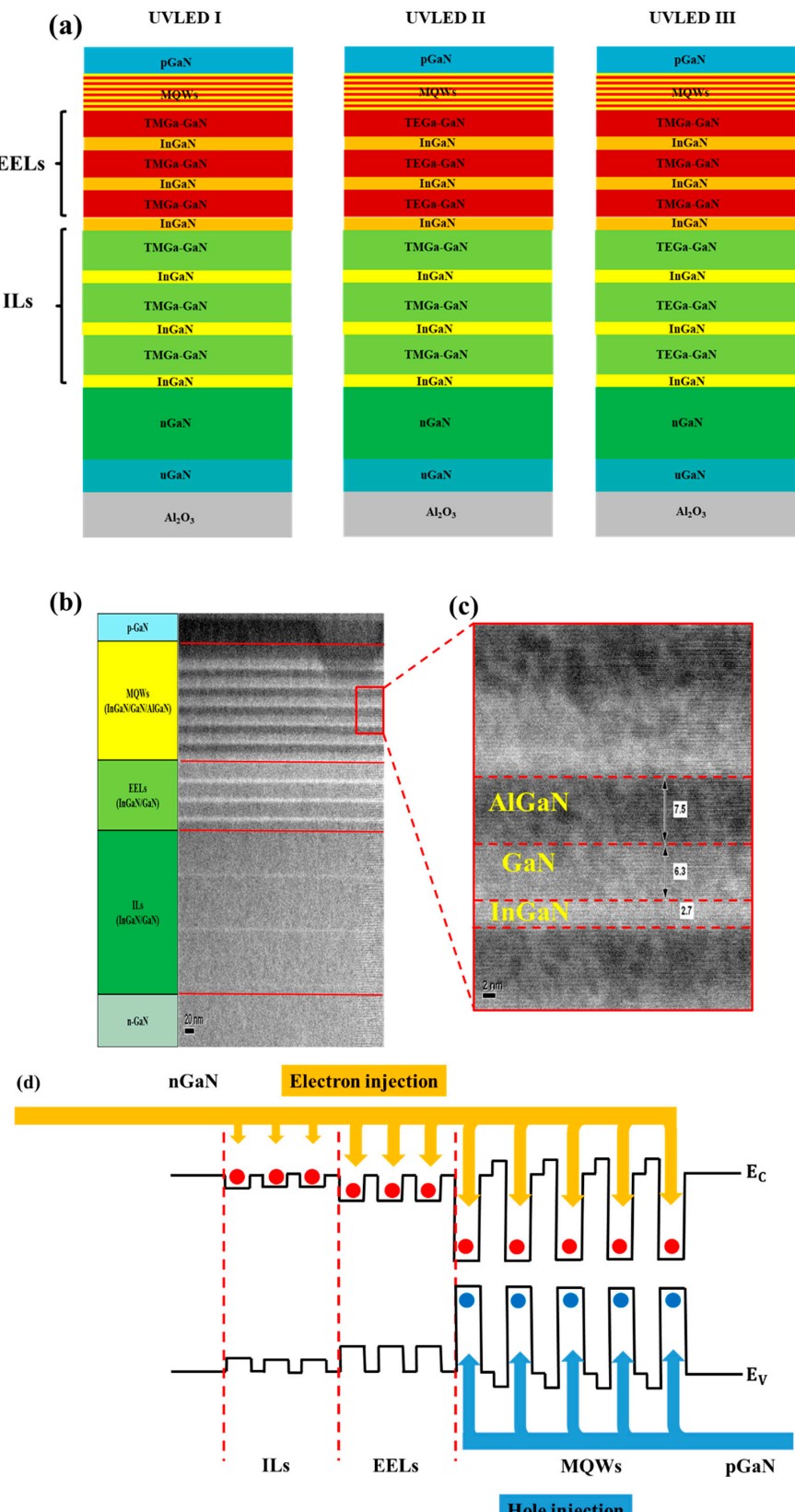

**Figure 1.** (**a**) Schematic diagrams of a full epi-structure with different growth conditions for electron emitting layers (EELs) and interlayers (ILs); (**b**) a cross-sectional TEM image of the UV-A light emitting diode (LED) structure; and (**c**) magnified cross-sectional TEM image for InGaN/GaN/AlGaN multiple quantum wells (MQWs). (**d**) Schematic band diagram of the UVLED structure.

## 3. Results and Discussion

Figure 1b shows the schematic diagrams and TEM cross-sectional image of the layer structures grown on the sapphire substrates by MOCVD. The epi-structures were composed of n-type GaN/un-doped GaN layer, three pairs of Si-doped 2-/6- nm thick $In_{0.01}Ga_{0.99}N$/GaN interlayers (IL), and three pairs of 3-/12-nm thick $In_{0.01}Ga_{0.99}N$/GaN EELs. The last six pairs of InGaN/GaN MQWs with an 8-nm thick AlGaN barrier are clearly observable in the TEM image (Figure 1c), as is a p-GaN layer with 150-nm thickness. Figure 1d illustrates the schematics of band structures and current/carrier flow of UVLED with large bandgap barriers. The effective barrier heights for both electrons in conduction bands and holes in valence bands are reduced, attributed to the overflowing. Thus, the thermionic carrier escape rate from the MQWs to the barrier layers increases, especially at a higher carrier/current injection. The insertion of EELs/ILs surrounding the MQWs results in an increased effective electron capture rate, which leads to thermionic carrier leakage suppression.

To evaluate the surface morphology of EELs before growth of the active layer, atomic force microcopy (AFM) measurements were carried out on a 5 μm × 5 μm size sample using the GaN barrier layer-terminated EELs/ILs/n-GaN/u-GaN on $Al_2O_3$ substrate (named following the UVLED chips as EI-UVLED I, EI-UVLED II, and EI-UVLED III) as shown Figure 2. The AFM images show well-defined step and terrace surface of the GaN barrier layer, which is related to terrace step-kink growth mode during the growth. In addition, a number of V-pits was also observed on the surface, which can affect the formation of pits or defects in the MQWs associated with a threading dislocation (TD) at the apex of a V-shaped pit with facets of {1011} due to the GaN growth process with the assistance of indium (In) atoms [13–15]. In addition, the observed etch pit density produced by propagation of TDs (yellow point) to the top surface of GaN in EI-UVLED I, EI-UVLED II, and EI-UVLED III is $1.7 \times 10^8$/cm$^2$, $1.6 \times 10^8$/cm$^2$, and $1.3 \times 10^8$/cm$^2$, respectively. The reduction of dislocation density in sample 3 is expected to show improvements in electrical and optical properties of UVLED devices. In addition, the surface roughness of the EELs/ILs/n-GaN/u-GaN/$Al_2O_3$ substrate of samples 1, 2, and 3 are 0.202, 0.242, and 0.210 nm, respectively. The varied surface roughness can be attributed to the different diffusion lengths of Ga adatoms with change of the MO source of TMGa and TEGa on the surface of GaN. The shorter diffusion length of Ga atoms from TEGa results in accumulation of Ga atoms at the kink, restricting lateral growth and producing a relatively rough surface morphology [16]. Moreover, the surface line profiles (red line) of the three samples along the terrace and step show the different lengths and heights, as displayed in the right column of Figure 2. The terrace widths and step heights of the GaN surface are clearly distinguishable with the change of the MO source for the growth of GaN barriers. The average terrace length and height were widened from 0.08 μm and 300 pm to 0.1 μm and 320 pm, respectively, with change of the MO source from TMGa to TEGa. EI-UVLED II showed intermediate terrace width and height values between those of EI-UVLED I and EI-UVLED III. The terrace width and step height of the GaN surface are important parameters to determine the growth mode of post grown epi-structures [17]. The step and terrace formations are related to the Ehrlich–Schwoebel barrier (ESB) occurring at the step-edge where the energy barrier is higher than the kinetic energy of the Ga adatoms on the terrace [18]. The ESB energies impede movement of Ga adatoms over a step-edge, which creates an asymmetry in the merge of adatoms transported from terraces. The ESB energy strongly affects the surface morphology. In practice, the ESB shows a stronger effect on step-edge at low growth temperatures [19]. When GaN layers are grown at the same temperature, TEGa is easily decomposed at a relatively lower growth temperature than TMGa. Therefore, the ES effect occurs easily in the GaN layer grown with TEGa compared to one grown with TMGa.

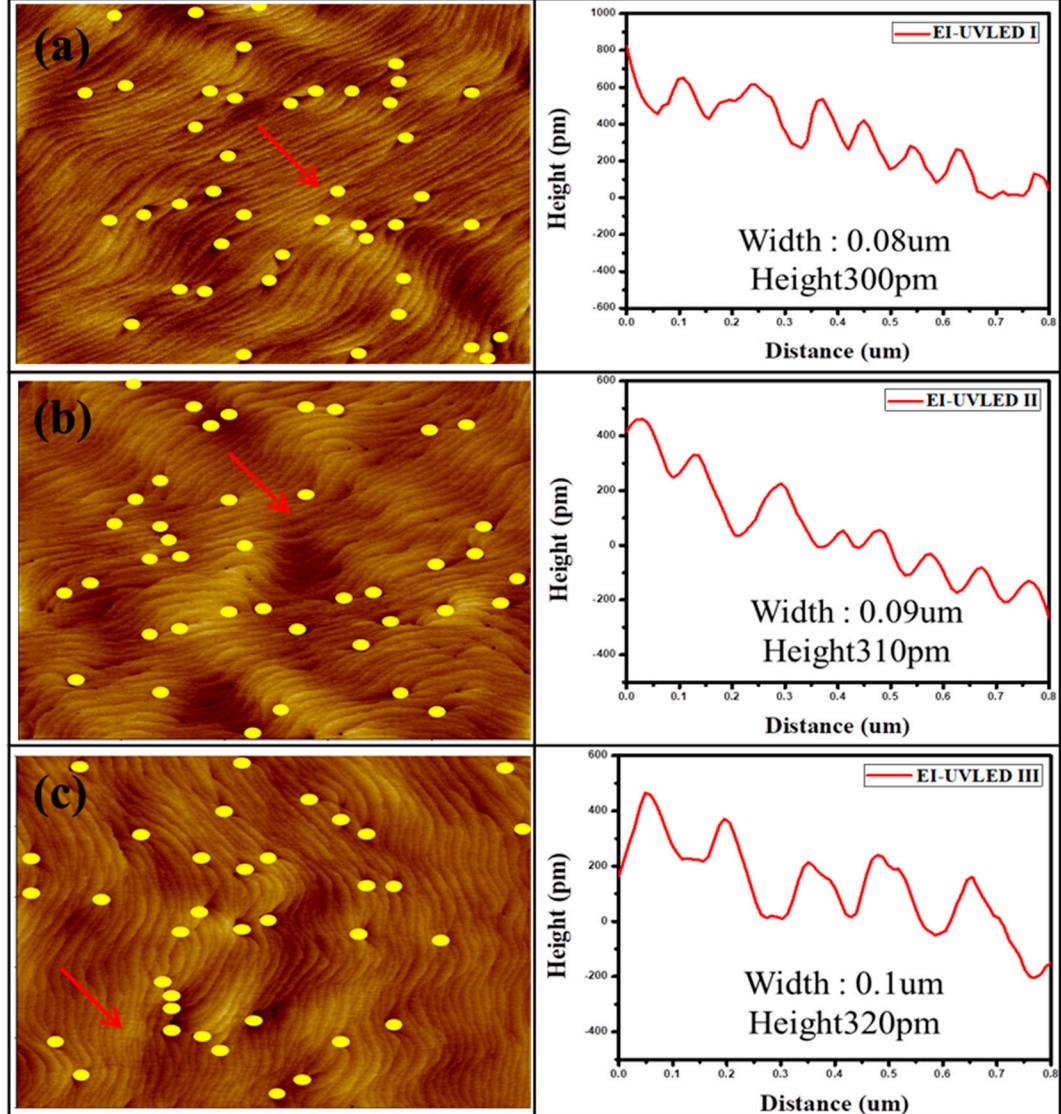

**Figure 2.** Atomic force microcopy (AFM) surface scan and line profiles of GaN/EELs/ILs for (**a**) EI-UVLED I, (**b**) EI-UVLED II, and (**c**) EI-UVLED III. Width and height are denoted in the figures.

To investigate the optical properties of the epi-structure grown on EELs/ILs/n-GaN/u-GaN on substrate, we measured the PL of three samples as shown in Figure 3. Two separate emissions at 360 nm and 365 nm are clearly distinguished for all samples and related to emission from GaN and InGaN, respectively. EI-UVLED III with GaN barriers grown by the TEGa source has a PL intensity 1.2 times higher than EI-UVLED I consisting of EELs/ILs with GaN barriers grown with the TMGa source. The improvement of PL intensity in EI-UVLED III was attributed to the optimized growth condition, due to the improved crystalline quality resulting from insertion of the GaN barrier with the TEGa layer.

For more detailed evaluations of the GaN barrier layer grown with different MO precursors, the GaN barrier layer was sequentially formed on the same u-GaN/sapphire template with a TMGa or TEGa MO source, and the PL properties of the grown GaN layers were compared as shown in Figure 4a. The GaN layer grown with TEGa has a near-band-edge PL intensity two times higher than the GaN layer grown with TMGa. The green to yellow emission PL peaks of the GaN layer represent the incorporated defects. The spectrum was converted to a logarithmic scale to confirm the defects related to the green and yellow emissions around 500 nm to 600 nm. The green to yellow emissions of GaN are dominated by deep level emission, showing point defects such as Ga-vacancy-related $V_{Ga}$–$O_N$

complex and edge dislocation density [20]. The integrated intensity ratios of the yellow luminescence band to the near-band-edge emission ($I_{YL}/I_{BE}$) are 0.04 and 0.16 for the GaN layer grown with TEGa or TMGa, respectively, as shown in Figure 4b. The deep acceptor-related emissions for $I_{YL}$ originated from the Ga-vacancy-related ($V_{Ga}$) native defects or its complexes with O and H in GaN thin films. Formation of point defects induced non radiative recombination centers that decreased PL intensity. Furthermore, the spectra show full width at half-maximum (FWHMs) peak emission of 8.2 and 11.6 nm at 360 nm of GaN grown by TEGa and TMGa, respectively. The PL analysis confirmed that the GaN layer grown with a TEGa source had better optical properties with relatively fewer defects than that grown with TMGa.

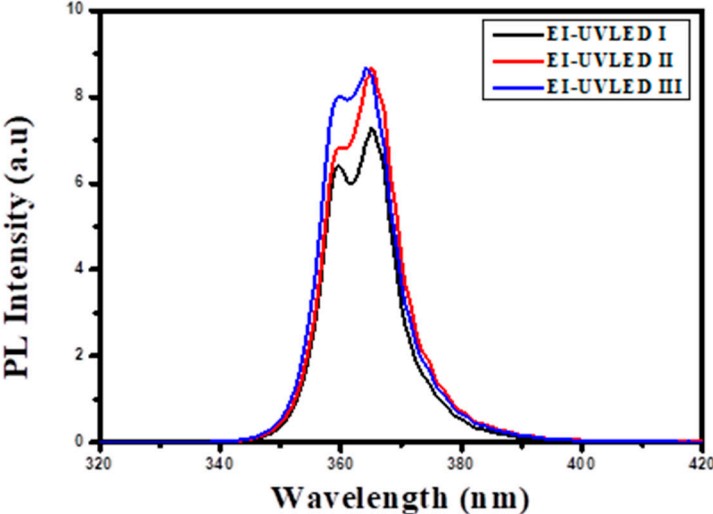

**Figure 3.** Photoluminescence (PL) spectra obtained from three samples with different GaN barrier growth conditions in EELs/ILs.

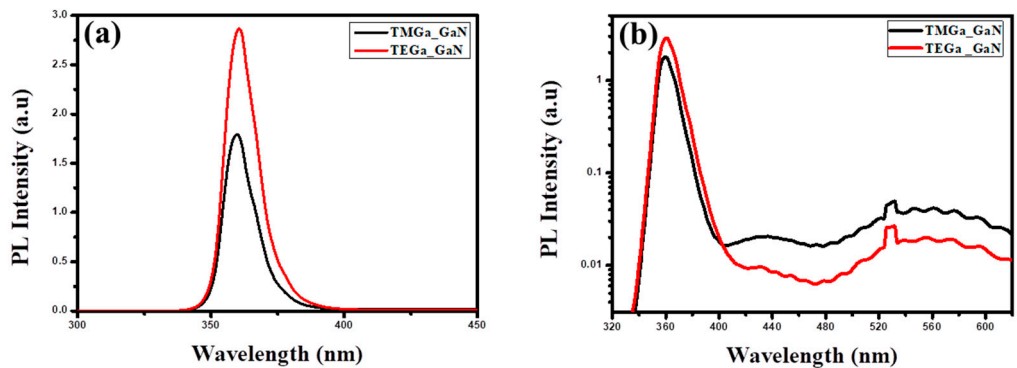

**Figure 4.** (**a**) PL spectra gathered from un-doped GaN grown with trimethylgallium (TMGa) and triethylgallium (TEGa) and (**b**) the spectra on a logarithmic scale.

Crystalline quality and interface of our epitaxial structures were evaluated by HR-XRD.

Figure 5a,b shows the HR-XRD rocking curves for the (002)- and (102)-plane reflections, respectively, obtained from the UVLED full structures grown on the $Al_2O_3$ substrates. The FWHMs of the (002) XRD rocking curve (XRC) of GaN with conditions of UVLED -I, -II, -III are 272, 266, and 250 arcsec, respectively. The FWHMs of the (102) rocking curve of the GaN layer with conditions of UVLED-I, -II, -III are 325, 310, and 285 arcsec, respectively. The FWHMs of rocking curves from epi-structures consisting of the GaN barriers grown with TEGa are narrower than those of the sample grown with TMGa, indicating superior crystal quality of GaN epi grown with TEGa. Generally, the rocking curve's FWHMs of (102) and (002) planes are very sensitive to edge and screw dislocation densities,

respectively [21]. The threading dislocation (TD) density can be calculated from the FWHMs of the XRD rocking curve (002) and (102) by Equation (1) [22].

$$N = \frac{\beta^2}{4.35 * |b|^2} \tag{1}$$

where N is the TD density, $|b|$ is the magnitude of the Burgers vector $\left(b_e = \frac{1}{3}\langle11\bar{2}0\rangle\right.$ and $b_s = \langle0001\rangle$, respectively$\left.\right)$ [23], and $\beta$ is the FWHM of the XRC. According to Equation (1), the density of screw dislocation in the GaN layer is $1.48 \times 10^8$, $1.42 \times 10^8$, and $1.25 \times 10^8$ cm$^{-2}$, and density of edge dislocation is $5.60 \times 10^8$, $5.09 \times 10^8$, and $4.31 \times 10^8$ cm$^{-2}$ for UVLED-I, -II and -III, respectively.

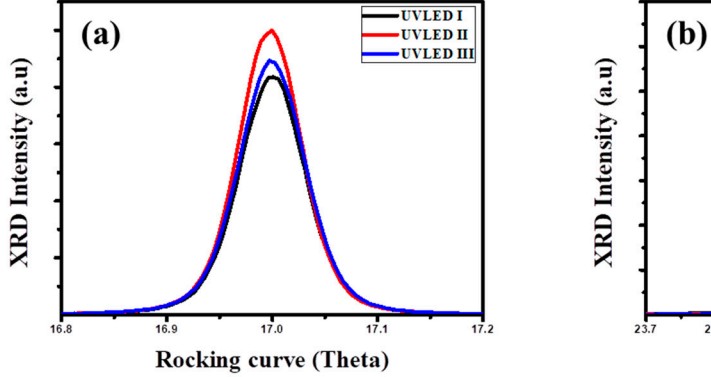 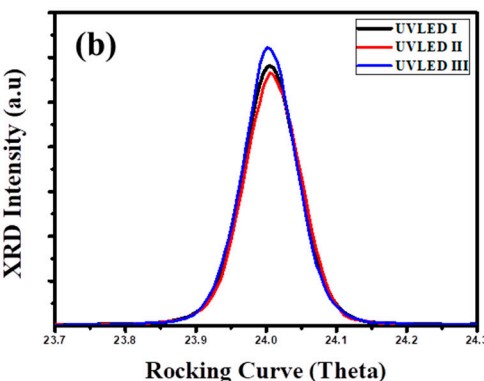

**Figure 5.** High-resolution X-ray diffraction (HR-XRD) rocking curves (**a**) of the (002) reflection and (**b**) the (102) reflections for the full structures UVLED-I, -II, and III.

After fabrication of the UVLED chip based on the material structures with InGaN/GaN/AlGaN MQWs grown on $Al_2O_3$ substrates, optical and electrical device performance was characterized. Figure 6a shows the current–voltage (I–V) curves measured from all of the fabricated UVLEDs in linear scales. The measured I−V characteristics show a voltage turn on at 3.19, 3.15, and 3.12 V for UVLED-I, II, and III, respectively. The lower forward voltage can be attributed to the low surface resistivity and low leakage current, forming a high current injection in the MQWs. The output powers of UVLED-I, II, and III are estimated to be 0.89, 1.04, and 1.21 mW, respectively, at a fixed injection current of 300 mA. As observed in Figure 6b, UVLED III shows 1.5 times greater light output power at 300 mA compared to UVLED-I. The inserts in the figure are the fabricated UVLED chip and light emission image at 300 mA injection current. Figure 6c shows the PL spectra of UVLEDs consisting of EELs and ILs with GaN barriers grown with TEGa and TMGa. The PL spectra were measured at room temperature using Nd:YAG laser ($\lambda$ = 266 nm) with laser source power of 2 mW. As shown in Figure 6c, the PL spectra intensity of the UVLEDs with the TEGa-grown EELs and ILs was much higher than that of the UVLEDs grown with TMGa. The integrated PL intensity of UVLED-III increased by 2.8 times compared with that of UVLED-I. In addition, Figure 6d shows a comparison of the electroluminescence (EL) intensity of the UVLEDs at an injection current of 300 mA. The light intensity increased by 1.9 and 2.3 times for UVLED-III compared to UVLED-II and I, respectively. This enhancement of PL and EL emission intensity can be attributed to the low impurity density and the high crystal quality of epitaxial layers resulting in higher light extraction efficiency in UVLEDs.

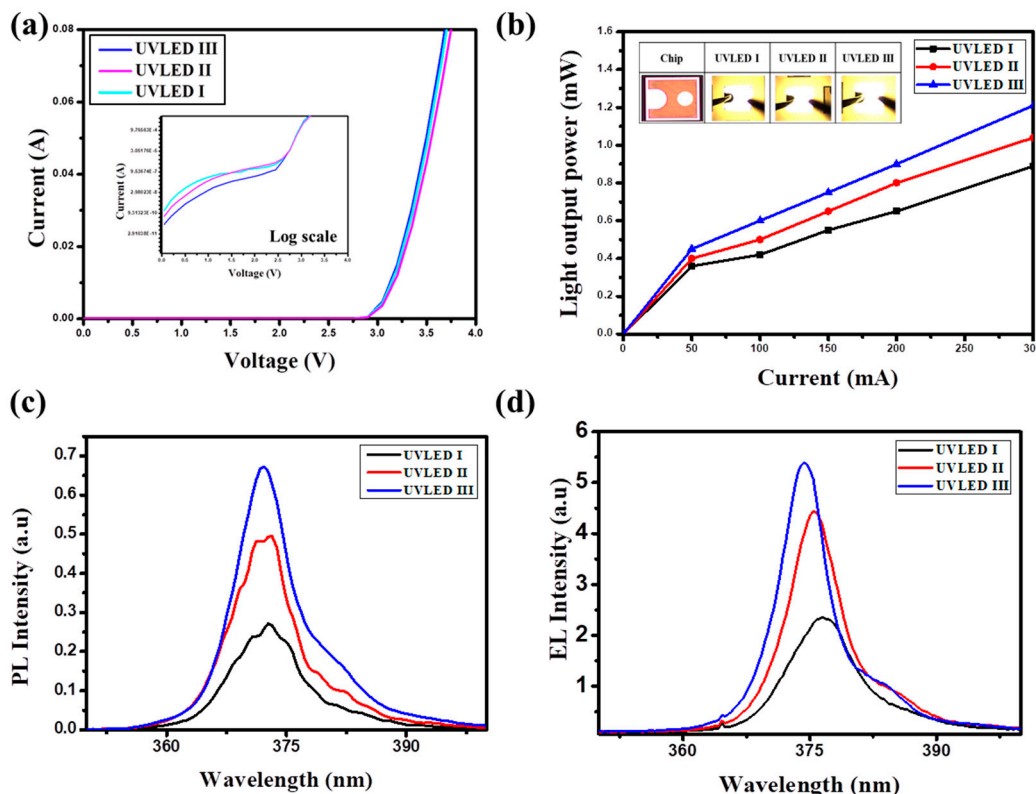

**Figure 6.** (**a**) Typical forward current–voltage (I–V) characteristics of UV-A LED (a semi-log plot is shown in the inset), (**b**) light output power as a function of injection current for three samples of UV-A LEDs, (**c**) PL, and (**d**) EL spectra of the UV-A LEDs.

## 4. Conclusions

We demonstrated the effects of InGaN/GaN-based ILs and EELs with different GaN barrier layers grown using TEGa and TMGa sources under the active layer. UVLED chips were fabricated and evaluated by light output power (LOP) measurements. In UVLED-III, the ILs and EELs grown with the TEGa MO source showed better epi-structural and electrical properties than those grown with the TMGa source. UVLED-III showed 1.5 times greater light output power at 300 mA current injection compared to UVLED-I. This indicates that insertion of EELs/ILs with a GaN barrier layer grown with TEGa is effective to improve device performances of UVLEDs. In addition, PL and AFM measurements clearly showed that the introduction of the GaN barrier layer with a TEGa MO source significantly improved the crystal quality of the UVLED structure. The growth of the GaN layer with TEGa increased growth time and had a slower growth rate compared with TMGa, resulting in reduction of dislocation density at the grain boundary. As a result, UVLEDs with ILs and EELs consisting of a GaN barrier layer grown with TEGa showed enhanced optical and electrical performances.

**Author Contributions:** J.P. and K.M.S. conceived and designed the experiments; D.K. and K.M.S. performed the growth and fabrication experiments; U.J., S.K., and D.S.S. conducted optical and electrical measurements and analyzed the data; J.P. and D.K. wrote the paper. All authors have read and agreed to the published version of the manuscript

**Funding:** This research was funded by the Ministry of Education, Science, and Technology (NRF-2018R1D1A1B07048382).

**Acknowledgments:** J. Park is grateful for the financial support from the Basic Science Research Program of the National Research Foundation of Korea (NRF).

**Conflicts of Interest:** The authors declare no conflicts of interest.

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
