# Peer review of "Effects of Different InGaN/GaN Electron Emission Layers/Interlayers on Performance of a UV-A LED"

_applsci, doi:10.3390/app10041514_

Round 1

Reviewer 1 Report

The manuscript reports on timely investigation of optoelectronic properties of InGaN/GaN structures prepared by various procedures. Presented results clearly demonstrate the improvement of the structure performance optimized as the sample No.3. The manuscript should be published in Applied Sciences after authors consider or amend few items listed below.

1. The investigated device shows pretty complex arrangement. I suggest authors to show the profile of band structure along the growth axis of their sample. Considered optical transitions might be sketched as well. I would help readers to better understand the topic.

2. The dimension units of micro-meters should be labeled correctly, not as um used for example in line 122.

3. Some typing or grammar faults were noticed, see for example in line 91: 'an n-type' and in line 210: 'a laser'. Fig. 4(a) 'PL Intensity'

Round 2

Reviewer 2 Report

The authors have addressed all the issues raised in my previous review in the revised manuscript. I am happy with the revision and the paper may be accepted for publication.